# Kidney Bean Fermented Broth Alleviates Hyperlipidemic by Regulating Serum Metabolites and Gut Microbiota Composition

**DOI:** 10.3390/nu14153202

**Published:** 2022-08-05

**Authors:** Weiqiao Pang, Di Wang, Zhaohang Zuo, Ying Wang, Wei Sun, Naidan Zhang, Dongjie Zhang

**Affiliations:** 1College of Food Science, Heilongjiang Bayi Agricultural University, Daqing 163319, China; 2National Coarse Cereals Engineering Research Center, Daqing 163319, China

**Keywords:** kidney bean fermented broth, hyperlipidemia, gut microbiota, untargeted metabolomics, serum lipid

## Abstract

Hyperlipidemia with fat accumulation and weight gain causes metabolic diseases and endangers human body health easily which is accompanied by metabolic abnormalities and intestinal flora disorders. In this study, the kidney bean fermented broth (KBF) was used in rats that were fed a high-fat diet to induce hyperlipidemia in order to subsequently analyse the serum metabolomics and gut microbiota modulatoration. The results show that the contents of the total polyphenols and total flavonoids in the KBF were up three and one times, while energy and carbohydrates decreased. In the HFD-induced hyperlipidemic model, body weight, organ weight, and the level of blood lipids (ALT, AST, TG, TC) were lower in rats treated with KBF than in the controls. Metabonomics indicate that there were significant differences in serum metabolomics between the KBF and the HFD. KBF could significantly improve the glycerophospholipids, taurine, and hypotaurine metabolism and amino acid metabolism of hyperlipidemic rats and then improve the symptoms of hypersterol and fat accumulation in rats. The relative abundance of beneficial bacteria increased while pathogenic bacteria decreased after the intervention of KBF. KBF ameliorates dyslipidemia of HFD-induced hyperlipidemic via modulating the blood metabolism and the intestinal microbiota. Collectively, these findings suggest that KBF could be developed as a functional food for anti-hyperlipidemia.

## 1. Introduction

Hyperlipidemia is a common chronic disease of lipid metabolism disorder in the body, which manifests as abnormal changes in the content of some lipid components, obesity, and fat content in organs [1,2]. Abnormal increase of TC, TG, and LDL-c could induce a series of diseases such as atherosclerosis, obesity, coronary heart disease, fatty liver, stroke, diabetes, hypertension, and sudden cardiac death [3,4]. Approximately 17.3 million people worldwide died from cardiovascular diseases in 2008, accounting for 48% of global deaths, and cardiovascular disease mortality is projected to reach 23.3 million by 2030 [5,6]. Recently, it has seriously troubled public health and aroused the attention of the public and researchers. Currently, drugs for treating hyperlipidemia, such as atorvastatin calcium, are expensive and have toxic side effects [7]. Therefore, the development of natural functional food with anti-lipid qualities has aroused people’s attention.

A number of studies have confirmed that the proportion of polyphenols that are enriched after fermentation intake should be increased in a reasonable diet structure to prevent hyperlipidemia, and the functional components play a potential hypolipidemia-lowering effect through different regulatory mechanisms [8,9]. For example, flavonoids in oats inhibit fat production and promote its decomposition [10]. Kidney beans are rich in nutrients such as protein and carbohydrates, rich in bioactive ingredients such as flavonoids, polyphenols, and dietary fiber [11,12,13] while providing people with abundant and reasonable dietary choices. Meanwhile, kidney beans have the effects of anti-oxidation, lowering blood sugar, and lowering blood lipids [14,15,16,17]. Fermentation with probiotics is an efficient, economical, and safe method of multigrain that decomposes macromolecular nutrients into small metabolites that can be easily absorbed [18,19]. It not only improves the flavor quality of multigrain but also enriches many functional bioactive ingredients. For example, the content of nutrients and the quality of oats increased after Lactobacillus plantarum and Rhizopus oryzae were fermented, and the active components such as polyphenols were enriched, and thus the antioxidant activity was improved [7]. However, it is not clear whether the liquid fermentation of probiotics can improve the active ingredients of kidney beans and the positive effects of improving dyslipidemia in vivo and antioxidant capacity. Meanwhile, fermentation liquid protecting against disease is linked to the regulation of the abundance of gut microbiota [20].

A growing number of studies have proved that there is a close relationship between the intestinal microbiome and hyperlipidemia. Long-term intake of a high-fat diet reduced the diversity of intestinal flora, the abundance of beneficial bacteria, and the ratio of Bacteroidetes to Firmicut [21]. Therefore, improving the structure of intestinal flora is considered a new target for the prevention and treatment of hyperlipidemia and increasing the abundance of probiotics to maintain a healthy and stable intestinal environment. Food fermented with probiotics could increase the diversity of intestinal flora, regulate the homeostasis of intestinal flora, participate in the regulation of lipid, glucose, and energy metabolism, and then reduce blood lipids, improve health, and inhibit the growth of harmful bacteria in the intestinal tract [22,23]. Current studies have shown that the content of flavonoids or polyphenols increased with probiotic fermentation by regulating intestinal microbial abundance to reduce hyperlipidemia induced by a high-fat diet [9,24]. Nevertheless, there are few reports on the mechanism of whether probiotics in fermented kidney bean liquid can regulate the intestinal microbial community structure in hyperlipidemia.

In order to maximize the intervention and adjuvant treatment of hyperlipidemia by probiotics from fermented cereals, the mechanism of probiotics from fermented cereals was studied from the perspective of intestinal flora. This study prepared a kidney bean fermentation liquid with probiotics and evaluated its effect on hyperlipidemic rats induced by a high-fat diet. Meanwhile, the apparent indexes, metabolites serum, and microbial community structure in gut microbiota were determined based on 16SrRNA gene sequencing. Finally, the correlation between apparent indicators, intestinal microbes, and serum metabolites was analyzed, aiming to provide a theoretical basis for the development of kidney beans and probiotic products or dietary supplements and to better understand the mechanism of probiotic fermented kidney beans reducing hyperlipidemia through regulating intestinal flora.

## 2. Materials and Methods

### 2.1. Materials and Samples

Kidney beans (violet-red speckled) were obtained from the Heilongjiang reclamation area (Heilongjiang, China). Angel yeast was purchased from Hubei Angel yeast Co., Ltd. (Yichang, Hubei, China). *Lactobacillus plantarum* and *Lactobacillus acidophilus* were stored in the library in strains in Northeast Agricultural University (Harbin, Heilongjiang, China). All ELISA kits were purchased from the Nanjing Jiancheng Institute of Bioengineering (Nanjing, Jiangsu, China).

### 2.2. Sample Preparation of KBF

The best fermentation condition was optimized: cleaned and peeled violet-red speckled kidney beans were smashed according to a 1:3 (g/mL) ratio of material to water. Subsequently, the kidney bean liquid was liquefied and saccharified through double enzymatic hydrolysis. The liquefying condition was 200 μL/100 mL of α-amylase, pH 6.0, 97 °C, 20 min. The saccharifying condition was 200 μL/100 mL of glucoamylase, pH 4.5, 60 °C, 30 min. Afterward, the kidney bean liquid supernatant was separated by centrifugal force at a condition of 1.0 × 10^4^ r/min, 15 min. They have been sterilized after adjusting pH = 4.0, 30 mL bottling quantity, and 6% sugar addition. Finally, 0.2% yeast was inoculated in a cool kidney bean liquid supernatant for pre-fermentation at 32 °C for 24 h. Subsequently, 1.5% *Lactobacillus plantarum* and *Lactobacillus acidophilus* were inoculated at 1:1 (co-fermentation) for 24 h.

### 2.3. Determination of the Basic Nutrient Content and Bioactive Components

The chemical composition of KBF and KBNF, including crude protein, lipid, ash, moisture, carbohydrate, starch, energy, soluble solids, total polyphenols, and total flavonoids were measured by National Engineering Coarse Grain Technology Research Center, Heilongjiang Bayi Agricultural University. All assays were performed in triplicate.

### 2.4. Animal and Diet

Thirty clean-grade SCXK (Liao) 2015-0001 SPF male Waster rats were obtained from Liaoning Changsheng Co., Ltd. (Liaoning, China), weighted (220 ± 20) g. All rats were housed in an SPF level animal cage at (23 ± 2) °C and relative humidity (50 ± 10)% with a 12/12 h light/dark cycle and free access to food and water in a specific pathogen-free experimental animal room in Heilongjiang Bayi Agricultural University. All rats’ body weights were monitored regularly, and blood bioinformatics were recorded and collected from the tail.

After the week of adaptation, six rats were randomly selected as CON with the standard diet and the rest of the rats were fed continuously with a high-fat diet. After 6 weeks, obese rats were randomly divided into four groups (*n* = 6 for each group): (a). MOD with saline extracts (concentration was 0.9%); (b). KBF-H with a kidney bean fermentation broth content of 1 × 10^10^ CFU·mL^−1^, (c). KBF-M with a kidney bean fermentation broth content of 1 × 10^7^ CFU·mL^−1^; (d). KBF-L with a kidney bean fermentation liquid content of 1 × 10^5^ CFU·mL^−1^. Intragastric administration was adopted regularly to feed rats in each group with a dose of 2 mL. The handling of the rats and the experimental protocol were also performed according to the Directive 2010/63/EU on the protection of animals used for scientific purposes (European Parliament and Council, Directive 2010/63/EU of 22 September 2010 on the protection of animals used for scientific purposes. Off. J. Eur. Union 2010, L276, 33–79.).

### 2.5. Measurement of the BW, Food Efficiency Ratio and Organ Weight

During the experiment, the body weights of the rats were measured after a 12 h overnight fast every week and the food intake of the rats was recorded. The food efficiency ratio was calculated as body weight gain/food intake. After the rats were killed and dissected, the liver, heart, kidney, spleen, pancreas, epididymal visceral adipose, and brown fat were removed. The residual blood on the surface was washed with saline and dried with filter paper. Lastly, the organ mass was weighed.

### 2.6. Serum Biochemical Analysis Hyperlipidemia

The rats were executed after fasting for 12 h, and the blood was quickly removed. Blood samples were centrifuged at 3000 rpm for 15 min and the serum was collected for the following analysis. Levels of serum lipids including TC, TG, HDL-C, LDL-C, ALT, and AST using an assay kit from Nanjing Jiancheng Bioengineering Co., Ltd. (Nanjing, Jiangsu, China). The arteriosclerosis index (AI) was calculated as the following Equation (1):(1)$$\mathrm{AI}=\frac{\mathrm{TC}-\mathrm{HDL}-C}{\mathrm{HDL}-C}$$

### 2.7. Metabolites in Serum Analysis

200 μL aliquots of serum were added to 800 μL methyl alcohol–water (1:4, *v*/*v*) and ultrasonic extracted (30 min, 40 KHz, 5 °C). The samples were incubated for 30 min (−20 °C) and finally centrifuged (13,000× *g*, 15 min, 4 °C). The supernatant was transferred to a fresh glass vial for analysis. The quality control (QC) samples were prepared with 20 µL supernatant by mixing up all the samples with an equal aliquot. All samples and QC samples were analyzed in parallel with the analytical instrument for UPLC-QTOF/MS analysis on UHPLC-Triple TOF system. Chromatographic separation was performed with an ACQUITY UPLC HSS T3 column (100 mm × 2.1 mm, 1.8 μm) at 40 °C (Waters, Milford, MA, USA). The flow rate was set at 0.40 mL min^−1^ and the injection volume was set at 10.0 μL. The flow blended with distilled water (A) and acetonitrile (B) for the positive and negative modes. Mobile phase A consisted of 95% of H_2_O and 5% of Methyl Cyanide. Mobile phase B consisted of 47.5% of Methyl Cyanide, 47.5% of isopropyl alcohol, and 5% of H_2_O. The identification of metabolites was performed in Progenesis QI (Waters Corporation, Milford, MA, USA) according to mass fragmentation (MS/MS analysis) and appropriate standards. At the same time, metabolites were identified according to the matching score of secondary mass spectrometry.

### 2.8. 16S rRNA-Amplicon-Based Sequencing and Microbial Analysis

Microbial genomic DNA was extracted from rat colonic feces using the QIAamp DNA stool mini kit for DNA according to the manufacturer’s instructions. The hypervariable V3-V4 region of the bacterial 16S rRNA gene was analyzed. All DNA samples were amplified and purified using primers pair (338F and 806R). Finally, the high-throughput sequencing was performed on the Illumina HiSeq platform by HiSeq2500 PE250 at Majorbio Bio-Pharm Technology Co., Ltd. (Shanghai, China).

### 2.9. Bioinformatics and Statistical Analyses

The original data measured on the computer were preprocessed to facilitate the screening and analysis of potential target differential metabolites. For metabolomics analysis, Progress QI (Waters, Milford, MA, USA.) was used for data processing, baseline filtering, peak identification, integration, retention time correction, and peak alignment. An accuracy error of 10 ppm was set in both MS and MS/MS data to confirm the tentative identification of metabolites. The available online databases, including HMDB (http://www.hmdb.ca, (accessed on 25 February 2022)) and METLIN (https://metlin.scripps.edu, (accessed on 25 February 2022)), were used to identify the chemical structure and name of metabolites through the obtained data. The resulting matrix was further reduced after removing peaks with the missing value in more than 80% of samples. Principle component analysis (PCA) and orthogonal partial least squares-discriminant analysis (OPLS-DA) were subsequently used to evaluate the overall difference between groups using the free online platform of Majorbio Cloud Platform (https://www.majorbio.com, (accessed on 25 February 2022)). The metabolic pathway was analyzed using the Kyoto Encyclopedia of Genes and Genomes (KEGG) database resource (https://www.kegg/jp, (accessed on 25 February 2022)).

The high-throughput sequencing was performed on the Illumina HiSeq platform by HiSeq2500 PE250 at Majorbio Bio-Pharm Technology Co., Ltd. (Shanghai, China). The original data obtained by sequencing were spliced and filtered to get effective data. Then the taxonomic analysis of operational taxonomic units (OTUs) aggregation and species was carried out using the RDP classifier Bayesian algorithm. The sequences were clustered into the same OTUs based on a similarity of greater than 97% (USEARCH, 1 http://drive5.com/uparse/, version 7.1.1090,(accessed on 25 February 2022)). According to the results of OTUs clustering, the representative sequences of each OTUs were annotated to obtain the corresponding microbial species information and the abundant distribution of microbial species. Alpha diversity was calculated by Mothur version v.1.30.2 (http://www.mothur.org/ (accessed on 25 February 2022)), including Chao1 richness estimator, Ace richness estimator to assess the information of species richness and Shannon-wiener, Simpson diversity index to assess the community diversity of microbe in the intestinal flora. The Rarefaction curve, Shannon diversity curve, and Rank abundance curve were also drawn. Principal component analysis (PCA) was used to evaluate beta diversity based on unweighted Unifrac distance. Significant differences in beta diversity were analyzed by QIIME 1.9.1 (http://www.mothur.org (accessed on 25 February 2022)). Linear discriminant analysis (LEfSe) was used to identify the most differentially abundant taxa of bacterial communities. The neighbor-joining method was used to construct the Phylogenetic tree by PyNAST (version1.2.2, http://biocore.github.io/pynast/ (accessed on 25 February 2022)) based on log-transformed OTUs. The highest number of the top 80 species was used to image a heat map by R Programming Language.

### 2.10. Statistical Analysis

All data were expressed as mean ± standard deviation (mean ± SD), except for gut microbiota. GraphPad Prism version 9.0 (GraphPad Software, San Diego, CA, USA) was used to study the differences among the different groups and transformed data to figure. Significant differences in groups were performed using Tukey’s test and one-way ANOVA at *p* < 0.05 level. The correlations between metabolites and microbiota were analyzed using the Spearman correlation analysis.

## 3. Results

### 3.1. Fermentation Changes the Bioactive Ingredients of KBF

Consumption of kidney beans is associated with beneficial effects on lipid metabolism [25]. The basic and bioactive components of the KBNF and KBF are shown in Table 1. The contents of protein (*p* < 0.05), ash, and starch (*p* < 0.01) decreased in the KBF compared to KBNF. This might be because the probiotic growth in the fermentation broth consumes some nutrients, making them less abundant. Additionally, carbohydrate, energy, and soluble solids decreased observably (*p* < 0.001) by 6.38 times, 2.5 times, and 5.05 times after fermentation, respectively. The bioactive compounds of total polyphenol were (30.02 ± 0.62) mg/mL adding up to (92.24 ± 2.14) mg/mL higher than those of the KBNF (*p* < 0.001), the multiple of growth was 1.56. Nutrients in fermentation broth were sufficient at the early stage of fermentation, and secondary metabolites were produced by the mass reproduction of microorganisms in fwhich organic acids and enzymes could dissolve phenols from kidney beans. The total flavonoids were obtained remarkably higher than those of the KBNF (*p* < 0.001), (367.75 ± 4.24) μg/mL, the multiple of growth was 3.07. The activity of microorganisms caused plant cells to burst and antioxidant substances in raw materials to exude and synthesize.

### 3.2. Effect of the KBF on BW, Food Intake and the Food Efficiency Ratio

Dietary habits and structure tend toward a high-fat diet, which also makes the symptoms of hyperlipidemia accompanied by body weight and body fat rate increase. In this study, high-fat diet-induced rats were used to evaluate the inhibitory effect of probiotic fermented kidney beans on hyperlipidemia. The weight of rats was recorded every two weeks, as shown in Appendix A. After the completion of the animal experiments, the experimental group fed with a high-fat diet was significantly higher than that of the CON group (*p* < 0.05, *p* < 0.01), indicating that the body weight of the experimental group was increased, and the body weight of the rats in each group was about 10.70–20.31 g. With the same feeding conditions, the body weight of the fermentation liquid dose group was significantly lower than that of the MOD group, showing that KBF had a certain inhibitory effect on the increase of body weight of hyperlipidemia rats. Compared with the CON group, there was no significant difference in food intake between the MOD group and the other group, indicating that the intervention of KBF could effectively control body weight gain without affecting food intake under the condition of a high-fat diet.

### 3.3. Effects of KBF on Body Length, Abdominal Circumference and Organ Quality of Rats

The rats’ body length and abdominal fat could intuitively present the state of high fat, which played an important role in pathogenesis and hidden dangers to the body [26]. The organ index also reflected the nutrient status and organ pathological changes of rats in each group. When the organ was affected by a high-fat diet, its index fluctuated and exceeded the normal value, and the quality changed. Body length, abdominal circumference, and organ indexes of the high-fat rats at 8 weeks were measured, as shown in Appendix A. Compared with the CON, the body length and waist circumference of rats were increased after high-fat diet intervention, but there was no significant difference among all groups (*p* > 0.05). Compared with MOD, the body length and waist circumference of rats in the experimental group after different doses of kidney bean fermentation liquid intervention decreased with an increased dose. Compared with normal diet rats, the weight of organs in the high-fat model group was significantly increased (*p* < 0.05, *p* < 0.01). Compared with MOD, the weights of the heart, liver, and pancreas in KBF-H were significantly decreased (*p* < 0.05), the kidney was observably extremely decreased (*p* < 0.01), and weight loss was 0.22–3.72 g. The weight of the spleen in KBF-M and KBF-H was also extremely significantly decreased (*p* < 0.01). The decreased amounts, respectively, were 0.36 g and 0.39 g. Meanwhile, epididymal fat and scapular fat in MOD were significantly increased (*p* < 0.01), with respective growth of 3.23 g and 0.34 g. Compared with MOD, two parts of fat content were extremely significantly decreased after different KBF (*p* < 0.01), which was about 2–3 fold. Above all, the results indicate that KBF could reduce viscera quality and different parts fat content of high-fat rats.

### 3.4. Effects of KBF on Lipid Levels in High-Fat Rats

Hyperlipidemia is caused by dyslipidemia, which mainly manifests as high TC, TG, and LDL-C levels or low HDL-C levels in serum [27]. These indexes are the main risk factors for chronic diseases which are used to evaluate abnormal lipid metabolism diseases. Compared with CON, TC, and TG, LDL-C levels were significantly increased in MOD and different dose kidney bean fermentation liquid groups, while the HDL-C level was significantly decreased (*p* < 0.01). Compared with MOD, KBF-H could prominently reduce TC, TG, and LDL-C levels, and increase HDL-C levels in hyperlipidemia rats (*p* < 0.05), which decreased, respectively, by 32.87%, 19.07%, and 21.42% and increased by 33.51% (Figure 1a–d). Above all, the results show that KBF intervention could reduce lipid metabolism disorder in hyperlipidemic rats, and the higher the content, the more obvious the reduction effect. Moreover, ALT and AST infiltrated into the blood and increased activity when the liver cell membrane was damaged. Therefore, ALT and AST levels in serum were used as an important indicator to evaluate the external injury of liver cells [28] and the degree of liver damage. Compared with the normal group, ALT and AST activities of the MOD and experimental groups were significantly increased (*p* < 0.01), and the increments were 74.71 U/L and 44.63 U/L, (Figure 1e,f) indicating the liver of rats in the high fat model was damaged. Compared with MOD, ALT and AST had a decreased gradient. The activities of ALT and AST dramatically decreased in the KBF-M and KBF-H (*p* < 0.05), and the reduced amounts were 22.62 U/L, 34.75 U/L, 22.31 U/L, and 33.25 U/L. However, AST activity and AI index (Figure 1g) in KBF-L decreased significantly (*p* < 0.05), and ALT had no significant change (*p* > 0.05). KBF had a protective effect on the liver, decreasing serum lipids and preventing or reducing the risk of diet-related cardiovascular diseases.

### 3.5. Effects of KBF on Serum Metabolites

Primary metabolites and secondary metabolites were identified by UHPLC-QTOF-MS/MS technology. A total of 436 metabolites were identified in the rat’s blood samples. PLS-DA models were established for positive and negative pattern analysis of normal feeding and high-fat rats (Figure 2a,b). The scores plot exhibited a clear differentiation between the CON and MOD indicating the disturbed metabolic pattern in diabetic rats. The values of R^2^Y were high, and the values of Q^2^ (Q^2^ = 0.934, Q^2^ = 0.947) were greater than 0.5, indicating a good model’s reliability and predictability. The above content could assess data quality and identify potential biomarkers showing high-fat diet had a significant effect on serum metabolites in rats. PCA was first performed to characterize the clustering feature between CON and KBF-H. As shown in PCA plots, a clear separation between the CON group and other groups was observed in positive and negative modes. Dose group and CON group developed in the same trends, indicating that the blood metabolites of experimental rats after different treatments were close to normal rats. KBF-M and KBF-H were closer to CON (Figure 2c,d), demonstrating that the serum metabolites of hyperlipidemic rats were gradually closer to normal rats after KBF intervention, and the serum pathological status of rats was significantly improved.

FC and VIP were further used to screen the differential metabolites (metabolites simultaneously met the FC > 2, VIP > 1, *p* < 0.05) (Appendix A). The results show that there were 138 differential metabolites in the comparison between CON and MOD, among which 49 metabolites were up-regulated and 67 metabolites were down-regulated. The regulated metabolites were mainly fatty acids and steroids and steroid derivatives, glycerophospholipids (adipic acid, taurodeoxycholic acid, fasciculol, etc.). Compared with MOD, the differential metabolites of KBF-L, KBF-M, and KBF-H were 80, 127, and 146 (Figure 3a–d). These results indicate that the changes of metabolites in KBF-treated rats tended to the normal group.

All the differential metabolites in different comparison groups were matched with the KEEG database to obtain the pathway information of metabolites and conduct enrichment analysis, and obtain the pathway enriched with more differential metabolites. The larger the bubble, the darker the color (Figure 3e–h). The main metabolites of CON and MOD were enriched in d-glutamine and d-glutamate metabolism, Taurine and hypotaurine metabolism, arginine biosynthesis and alanine, and aspartate and glutamate metabolism. Compared with the MOD group, the main differential metabolite enrichment pathways of KBF-L were alanine, aspartate, glutamate metabolism, and histidine metabolism. However, the KBF-M metabolic pathway had a uniform bubble size and five metabolite enrichment pathways. Meanwhile, compared with MOD, the metabolite concentration in the pathway of KBF-H tended toward CON. These metabolic pathways overlap suggesting that KBF could effectively improve the amino acid metabolism pathway, taurine, and hypotaurine metabolism, and then enhance the serum metabolism of high-fat diet rats to a normal state.

### 3.6. Effect of the KBF on Regulation of Gut Microbiota

Previous studies have shown that the higher the gut microbiota diversity, the lower the individual susceptibility to disease [29]. Multiple different alpha diversity index analyses could reflect the community diversity of gut microbiota which is used to investigate the evenness and richness. The Chao index, ACE index, Shannon index, and Simpson index were calculated (Figure 4). Compared with CON, the differences between the Chao index, Shannon index, and Simpson index in MOD were significant (*p* < 0.05, *p* < 0.01). There were significant differences in richness indices (Chao and ACE) between KBF-H and MOD (*p* < 0.05). It indicated that the community of diversity of gut microbiota in rats decreased significantly (*p* < 0.05). The Shannon index significantly increased, and the Simpson index significantly decreased compared with that of the MOD group (*p* < 0.05). The Shannon index in KBF-H was higher than that in the MOD group. In contrast, the Simpson index was lower (Figure 3a,c). The result shows that kidney bean fermentation broth could significantly improve the community richness of hyperlipidemic rats. The intervention of KBF could increase and restore the bacterium diversity; however, it had no effect on the community’s richness.

A Venn diagram was constructed to analyze the overlap and similarity of OTUs’ composition in different groups in order to explore the effect of kidney bean fermentation broth on the gut microbiota of hyperlipidemic rats. As seen in Figure 5a, a total of the same 567 OTUs in each group indicated that there was core microbiota among the groups. Moreover, the unique number of OTUs of the CON, MOD, KBF-L, KBF-M, and KBF-H groups were 21, 6, 12, 12, and 13, respectively. The results indicate that kidney bean fermentation broth shaped the composition of the gut microbiota in rats. In order to further research the similarities and differences in the structure of fecal microflora of rats in each group, a heatmap was constructed to analyze the distribution of microflora at the phylum level with different treatments (Figure 5b). There were differences in the composition of the predominant phylum between CON and MOD (TOP 20 phylum). However, the composition of the predominant phylum in KBF-H was similar to CON. Based on the significantly changed gut microbiota induced by hyperlipidemia, we found that kidney bean fermentation liquid had a positive regulatory effect.

In the present study, the bacteria with the highest relative abundance in each experimental group were Firmicutes and Bacteroidote. Research has claimed that the ratio of *Firmicutes* to *Bacteroidote* could reflect the disturbance of gut microbiota induced by fatty and sugary foods [30]. Compared with CON, MOD obviously increased the relative abundance of Firmicutes, reduced the relative abundance, and, of course, heightened the ratio of Firmicutes/Bacteroidetes in the hyperlipidemia rats (Figure 5c, *p* < 0.01). After the KBF was administered, the relative abundance of *Firmicutes* was clearly reduced (*p* < 0.01), and the relative abundance of *Bacteroidetes* was slightly increased, compared with the MOD group.

Figure 5d shows the distribution of gut microbiota in rats at the genus level. After the intervention of high dose kidney bean fermentation broth, the relative abundance of *norank_f_Muribaculaceae* (41.15%), *Allobaculum* (3.16%), *Romboutsia* (2.04%), *norank_f_Ruminococcaceae* (2.36%), and *norank_f_Lachnospiraceae* (1.85%) was significantly higher than that of MOD group. Obviously, kidney bean fermentation broth could increase and reshape the diversity of beneficial microorganisms in the gut microbiota of rats and improve hyperlipidemia rats by regulating microflora. Additionally, different changes in gut microbiota composition between other groups and the CON group were analyzed on unweighted Unifrac at the phyla level.

The first principal coordinate (PC1) in the PCoA analysis explained 62.45% of the overall variation. The PC2 axis explained 18.61% of the variability in microbial communities. Unifrac distance and ANOSIM revealed a significant difference in the structure (Figure 6a, r = 0.2964, *p* = 0.001) of gut microbiota among different doses of kidney bean fermentation broth from the CON group, indicating that different treatments had significant effects on the gut microbiota of hyperlipidemic rats. In addition, compared with the CON group, the KBF-H was relatively coherent, proving that the intervention of kidney bean fermentation broth changed gut microbiota and had positive effects on relieving hyperlipidemia (Figure 6b).

A Wilcoxon rank sum test diagram was drawn to further investigate the microbiota with significant differences of relative abundance between diabetic rats and normal rats. At the genus level, there were significant differences in nine genera, including *Bifidobacterium*, *Escherichia-Shigella*, *Enterococcus,* and *Allobaculum* between the MOD group and KBF-H group (Pfdr < 0.05). In total, six genera were significantly enriched in the MOD group. Hyperglycemia is an important feature of diabetes and the root cause of infection. Like common pathogenic bacteria, *Escherichia coli* and *Shigella* easily grow in high-concentration glucose and cause a series of complications such as skin infection in diabetes patients. There were significant differences in the composition of the community between KBF-H and the CON group. The relative abundance of gut microbiota pathogenic bacteria degraded prominently by KBF treatment. LEfSe analysis was used to analyze the key phylotypes of gut microbiota among different groups at the genus level (Figure 6c,d). The results show that *Lactobacillus*, *Bifidobacterium*, *Escherichia-Shigella*, *Ruminococcus*, *Christensenellaceae*, *Enterococcus*, et al. were enriched in the hyperlipidemia rats compared to the normal healthy rats (LDA > 2, *p* < 0.05). *Lachnospirales*, *Clostridia_UCG-014*, *Ruminococcaceae*, *Lachnospiraceae*, *Saccharimonas*, *Anaerostipes*, *Desulfovibrionaceae*, *Monoglobaceae*, *Colidextribacter* were remarkably different compositions of gut microbiota in KBF-H group (LDA > 2, *p* < 0.05). Meanwhile, the relative abundance of gut microbiota pathogenic bacteria were degraded prominently such as *Escherichiacoli* and *Shigella* in the gut of hyperlipidemia rats. For example, the abundance of *Ruminococcaceae* and *Lachnospiraceae*, et al., and beneficial bacteria were enriched actively by a high dose of KBF. All results show that gut microbiota dysbiosis in hyperlipidemic rats induced by a fat diet was modulated and restored after treatment with KBF.

### 3.7. Association between the Serum Apparent Data, Metabolites and Gut Microbiota

The correlations between serum lipids apparent indicators, serum metabolites, and gut microbiota (based on the dominant results) were analyzed using Spearman’s correlation coefficient (Figure 7). Figure 7a shows that KBF intervention serum lipids apparent indicators were remarkably related to the gut microbiota. For weight, TC, AST, and ALT, body length was positively correlated with *Proteobacteria* (at the phylum). TC and TG were inversely correlated with *Bacteroidota* (*p* < 0.01). *Cyanobacteria* were positively correlated with TG, TC, and LDL-C. The correlation results indicate that serum fat and gut microbiota had a relative relationship of mutual influence. Moreover, the serum lipids changed by KBF might alter and improve the gut microbiota to alleviate hyperlipidemia (Figure 7b). Glycodeoxycholic acid and taurodeoxycholic acid show a positive correlation with weight, TG, TC, LDL-C, AST, and ALT; however, they showed a negative correlation with HDL-C (*p* < 0.01). Uric acid and d-Glucuronic acid were positively associated with HDL-C. Moreover, they were inversely associated with weight, TC, ALT, and AST. Indoxylsulfuric acid showed a positive correlation index for *Bacteroidota* and a negative correlation for *Actinobacteriota*, *unclassified_k_norank_d_Bacteria*, and *Proteobacteria*. Uric acid and d-Glucuronic acid were correlated with *Patescibacteria* and *Spirochaetota* (Figure 7c). Together, the correlation results indicate that serum lipids’ apparent indicators, metabolites, and gut microbiota constitute a triangular relationship of mutual influence in the hyperlipidemic model rats.

## 4. Discussion

Kidney beans are rich in nutrients (protein, phenolic compounds, etc.) that provide a good substrate for probiotics. More studies have shown that [31,32] hyperlipidemia has an intimate correlation with intestinal flora, and they affect and interact with each other. By supplementing probiotics and probiotic fermented food, the structure and diversity of intestinal flora could be significantly improved, thus they play a role in the treatment and prevention of hyperlipidemia. The potential of probiotic fermentation of plant-based (kidney beans) to reduce hyperlipidemia and the role of intestinal microflora and metabolites (in colon and serum) has been not clear. It is, therefore, necessary to explore the mechanism of kidney bean fermentation improving lipid and obesity in rats. Our results show that the functional components (polyphenol, flavone) were enhanced by KBF. Previous studies have also proved fermented grains could enrich certain functional components after fermentation, such as barley or black barley with probiotics and kidney beans fermented with *Pleurotus ostreatus* [9,33]. This may be because probiotic fermentation promotes the release of polyphenols and flavonoids in raw kidney beans. The content of carbohydrates increased in rice and black barley fermented with probiotics, while they were reduced in this experiment. The above beneficial effects might be partially attributed to certain fermented sugar as a source of carbon growth and reduction in the carbohydrate content [34]. Therefore, KBF will become a suitable drink for people with special needs.

Long-term HFD will affect abnormal changes in body weight with accumulating visceral fat and triglycerides and cause cardiovascular disease. Accumulating evidence has suggested that *Lactobacillus plantarum* fermentation of barley extraction could inhibit body weight and fat content increase [34], consistent with our findings. In this study, compared with a model group, the body weight and viscera quality of HFD rats treated with KBF decreased after 8 weeks, indicating that probiotic fermented kidney beans could effectively inhibit body weight gain and reduce body fat weight. Neil has proved that reduced consumption of white kidney beans reduced fat accumulation in obese mice because of higher bile acid and a lower ratio of *Firmicutes* to *Bacteroidetes* [13]. Hyperlipidemia is a chronic disease caused by abnormal lipoprotein metabolism, mainly including increased content of TC, TG, and LDL-c or a decrease in the content of HDL-c. Because of the abnormal elevation of blood lipids, the metabolism of lipids is regulated by a variety of enzymes in the body which play a key role in lipid metabolism or limiting the rate of metabolic reaction. Studies have shown that the RS of roasted red kidney beans could restrain the formation of cholesterol, reduce cholesterol levels, and relieve the abnormal lipid metabolism caused by a high-fat diet [32,35]. Compared with the high-fat model group, the serum level of TC, TG, and LDL-c in KBF-H were significantly decreased, and the levels of HDL-c were increased, because the functional components in KBF of probiotics activated key enzymes in TC and TG metabolic pathways, and further regulate lipid metabolism to achieve a hypolipidemic effect. These apparent index trends were consistent with probiotics fermented rice buckwheat and dietary millet whole grains [36,37]. Simultaneously, these results suggest that the long-term intake of whole grains could meet the nutritional and health demands of special populations to eat more a reasonable and healthy diet and reduce the incidence of chronic diseases. Untargeted analysis of metabolites in the blood is widely applied in recent qualitative and quantitative analyses of the metabolite enrichment of fermented food [38]. Studies have found that taurine was recognized as an essential nutrient in the body and had physiological and biochemical effects. The increase in taurine level was also closely related to the occurrence and development of obesity and reduced the TG level in the liver [39]. The results have shown that *Monascus* fermented buckwheat inhibits bile acid accumulation by reducing taurine levels [40]. Our study showed that kidney bean fermented broth could improve the physiological status of blood lipids by down-regulating the contents of valine, isoleucine, tyrosine, and taurine in high-fat rats, consistent with the *Monascus* fermented buckwheat. Previous studies have shown that uridine is a pyrimidine nucleoside that can affect liver energy metabolism and exert a protective effect against liver lipid accumulation [41]. The content of this substance was up-regulated after KBF intervention in this study, which was consistent with the results of the type and content change trend of metabolites affected by *Lactobacillus* fermentation of black barley on fatty liver metabolism [9]. Arginase acts on L-arginine to produce ornithine, thus forming polyamine, putrescine, spermidine, spermine, and L-proline [42]. The content of L-arginine in KBF-H was significantly higher than that in the MOD group. After KBF intervention, the metabolic pathways were mostly focused on Glutamine and Glutamate metabolic pathways, which also happened to coincide with the results of differential metabolism. We will analyze the metabolic pathway of fat from the perspective of lipid metabolomics, and then study the mechanism of KBF inhibiting fat formation in the follow-up study.

Further, we investigated whether the beneficial effects of KBF on ameliorating hyperlipidemia were connected with changes in the intestinal microbiota. The composition and function of intestinal flora are related to the energy metabolism and health of the host. The intestinal microecological disorder caused by the imbalance of intestinal flora is an important factor in the occurrence and development of hyperlipidemia. At present, the reports have demonstrated that consumption of red kidney beans, white kidney beans, and navy kidney beans could improve hyperlipidemia and obesity rat intestinal health and the integrity of intestinal epithelial cells, and alleviate the degree of obesity [21,43]. However, there have been few reports about fatty intestinal flora intervention with liquid fermented kidney beans. Therefore, the relationship results were analyzed by referring to other plant-fermented foods affecting the intestinal steady state in this study. *Allobaculum* has a potential physiological benefit for hosts [44]. Past research has shown that the relative abundance of *Allobaculum* in HFD mice increased significantly after the probiotic fermentation millet intervention [45]. In addition, there was a report that the abundance of *Allobaculum* was low in HFD mice and increased after the intervention of fermented lycium ruthenicum [44]. These results coincide with the results of KBF in the high-dose group in this paper. There was a significant difference of intestinal microflora of rats in CON and MOD. At the phylum level, the relative abundance ratio (F/B) of *Firmicutes* and *Bacteroidetes* in MOD was clearly higher than that in CON. The results of fermented balsam pear polysaccharide and barley [46] were consistent, and the F/B ratio was found to be significantly correlated with blood glucose and hyperlipidaemia. Previous studies have shown that the administration of rutin or other bioactive ingredients could reduce the F/B ratio, thereby improving lipid metabolism in HFD-induced mice [41]. Rice buckwheat fermentation more effectively reduces the F/B ratio [24]. These results suggest that fermented plant-based foods play a beneficial role by regulating the proportion and diversity of gut microbiota. *Bacteroidetes* constituted the main component of normal gastrointestinal flora. The research showed that the abundance of *Bacteroidetes* increased in mice fed with a standard normal diet, while decreased in mice fed with a high-fat diet, in agreement with this study [44]. It also agreed with TC, TG negative correlation, and its abundance increased after KBF intervention, indicating that it improved the intestinal microflora structure of rats. Compared with CON, *Proteobacteria*, *Escherichia*, *Shigella*, *Desulfovibrio,* and other gram-negative bacteria were significantly enriched in the MOD, and cytoderm component endotoxin could inhibit intestinal barrier function and absorb more LPS function. LPS forms a complex with the glucose-phosphoisomerase anchor protein CD14 and is recognized by TLR-4 on the surface of mononuclear macrophages, acting on skeletal muscle cells and adipocytes [47]. Meanwhile, LPS is also transferred to metabolic tissues such as the liver and pancreas, promoting the occurrence of metabolic endotoxia and systemic IR. *Desulfovibrio* is the main sulfate-reducing bacteria in the human gut that could reduce sulfate generation and generate H_2_S, thus damaging intestinal epithelial cells [48]. Research has shown that probiotic-fermented rice-buckwheat reduced harmful bacteria in the gut, such as *Desulfovibrio* and *Pseudomonas*, which may be related to the flavonoid and polyphenols in rice and buckwheat [24]. *Enterococcus*, *Escherichia*, and other symbiotic bacteria have certain potential hazards in the environment of intestinal internal disorder. Diabetes easily causes bacteria to enter other tissues and organs and secretes gelatinase due to its low immunity Cytolysin and other virulence factors invade, destroy host tissue cells, and tolerate the host’s non-specific immune response, increasing the risk of infection of the host’s abdominal cavity infection, blood sepsis, endocarditis, and other infectious diseases. The abundance of *Lactobacillus* was relatively low in HFD gut flora. It was added after intervening with probiotic-fermented rice-buckwheat and monascus-fermented buckwheat [9,24]. The relative abundance of *Lactobacillus* was consistent with this research; however, the high-dose group of KBF decreased because of the increase of other dominant bacteria, thus inhibiting the decrease of *Lactobacillus*. There are relatively few studies on the abundance of *norank_f_Muribaculaceae*. It improved the structure of intestinal flora and abundance in alcohol-drinking rats after Pu’erh tea intervention, which agreed with the changing trend of this study [49]. *Clostridium* has been proved to be an important bacterium for the formation of amino acids [44,50] and has an effect on the amino acid content in intestinal tissues, liver, and other tissues. *Clostridium* was also detected in the intestines of rats after the high-dose intervention, indicating that kidney bean fermentation broth could enrich the amino acid content in the viscera of rats in this study. The role of specific bacteria related to hyperlipidaemia in regulating hyperlipidaemia could be further studied through simulated in vitro fermentation experiments and metagenomics in future experiments.

Spearman correlation analysis showed that there were different degrees of correlation between blood apparent indexes, metabolites, and intestinal microflora in rats after KBF intervention. Previous studies show that *Bacteroides* are negative with hyperlipidemia-related parameters [21,40]. Consistent with these findings, we observed negative correlations between *Bacteroides* and TG, TC. Studies have shown that *Bacteroides* mainly contributed to the degradation of specific carbohydrates, such as plant polysaccharides (hemicellulose and pectin), host polysaccharides, and α-glucan, thus providing beneficial effects on glucose metabolism and energy absorption [51]. Therefore, KBF may have a sufficient impact on glycometabolism and energy absorption through upregulating the *Bacteroides*. *Blautia* has a remarkably negative relationship with physiological dysfunctions of the host, such as obesity, hyperlipidemia, and various inflammatory diseases [45]. No correlation between *Blautia* and hyperlipidemia was found in this study; however, the abundance of *Blautia* was increased after KBF intervention, indicating beneficial bacteria for hyperlipidemia. Research has confirmed that *Verrucomicrobia* was negatively correlated with TC [52], whereas this apparent correlation was not reflected in this experiment. However, there was a significant correlation between *Verrucomicrobia* and PC (phosphatidylcholine), which might indicate that *Verrucomicrobia* could reduce cholesterol levels and regulate serum lipid levels. Research has confirmed that *Patescibacteria* were highly correlated with ALD, and its abundance was high in the model group without the intervention of fermented sea buckthorn juice. The abundance of *Patescibacteria* showed a significant positive correlation with Uric acid, p-Tolyl Sulfate, and D-glucuronic Acid in our study. These metabolites exist in the liver mostly in patients with chronic diseases such as CKD, which also indicates that *Patescibacteria* are harmful bacteria in the gut. These results confirmed that dyslipidemia harms health and causes other chronic diseases.

There was a close correlation between metabolites and apparent data in this research. The results show that LysoPC could promote cholesterol efflux, prevent fatty acid synthesis, promote cholesterol oxidation into cholate, and reduce the level of TC, TG, and other indicators. The metabolites LysoPC (17:0) and LysoPE (22:0) in KBF had negative correlations with TC, TG, LDL-C, ALT, and AST, and positive correlations with HDL-C, suggesting that KBF could reduce fat formation in this study. These results correlated with monascus-fermented buckwheat [24]. Furthermore, according to the correlation analysis, KBF affects and regulates lipid levels in hyperlipidemic rats through metabolic pathways such as glutathione metabolism, sphingolipid metabolism, citric acid cycle, and glycerophospholipid metabolism. The intervention of KBF could regulate the transformation of metabolites and intestinal flora and then improve hyperlipidemia. Meanwhile, the improvement effects of KBF on lowering blood lipids could be determined by monitoring the changes in blood lipids and intestinal flora in hyperlipidemia.

## 5. Conclusions

In summary, the probiotic fermentation of kidney beans increased total polyphenol and total flavonoid, but decreased energy and carbohydrates, et al. The intervention of KBF improved abnormal blood lipids, lipid metabolism, and intestinal flora in HFD-induced hypolipidemic rats. Moreover, an effective dosage of 1 × 10^10^ CFU·mL^−1^ of KBF treatment was superior to low doses of KBF. Our results indicate that KBF showed its promising lipid-lowering effects by regulating lipid beneficial metabolites, key metabolic pathways in hyperlipidemia, and intestinal species richness and diversity. Correlation studies show that the apparent data, serum metabolites, and intestinal microbes had significant positive and negative correlations against obesity and hyperlipemia. These results indicate that KBF may potentially be used as an effective, safe, and affordable functional ingredient for the management of hyperlipidemia and obesity. However, further qualitative and quantitative analyses are required for the anti-adipogenic bioactive compounds detected in kidney bean fermentation broth, to illustrate the components of the correlation and molecular mechanism in improving the effect on hyperlipidemia.

## Figures and Tables

**Figure 1 nutrients-14-03202-f001:**
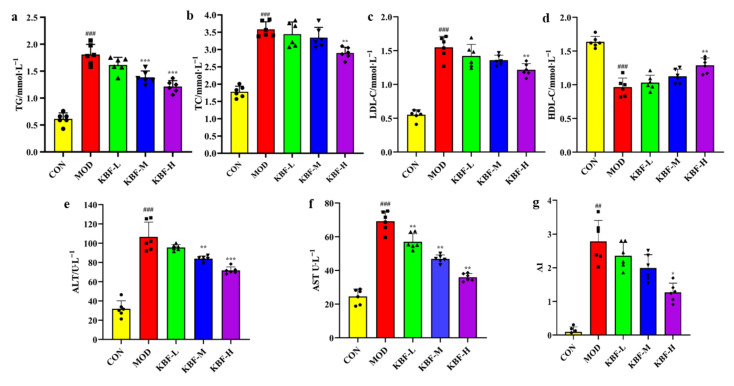
KBF on blood lipid in hyperlipemia rats. (**a**) Level of serum lipids including TC; (**b**) TG; (**c**) LDL-C; (**d**) HDL-C; (**e**) ALT; (**f**) AST; (**g**) AI. Data are expressed as the mean ± SD (*n* = 6). (^##^
*p* < 0.01, and ^###^
*p* < 0.001 versus CON), (* *p* < 0.05, ** *p* < 0.01, and *** *p* < 0.001 versus MOD).

**Figure 2 nutrients-14-03202-f002:**
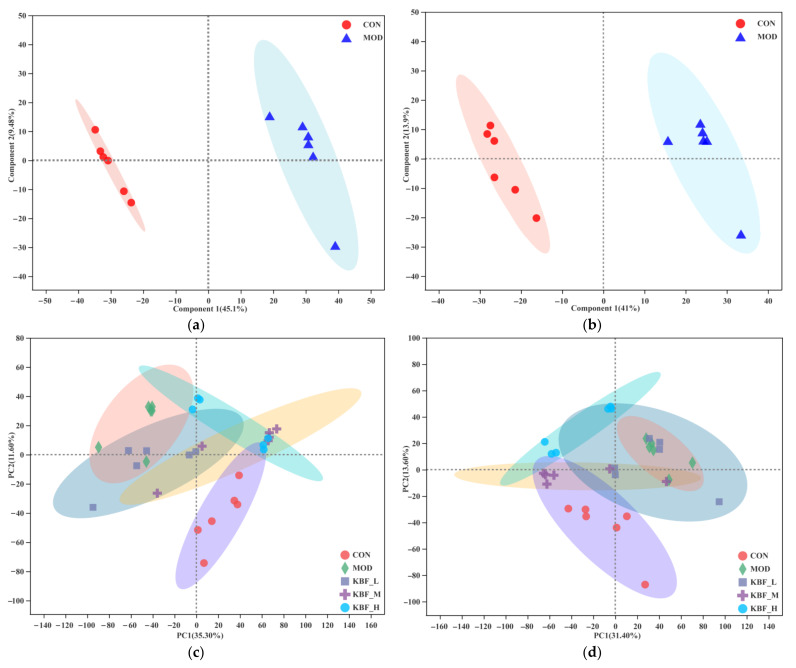
PLS-DA score plots between model and normal groups in the negative modes ((**a**), R^2^ = 0.964, Q^2^ = 0.934); positive modes ((**b**), R^2^ = 0.973, Q^2^ = 0.947); PCA score plots for the MOD, CON, KBF-L, KBF-M, and KBF-H groups in positive (**c**) and negative (**d**) modes.

**Figure 3 nutrients-14-03202-f003:**
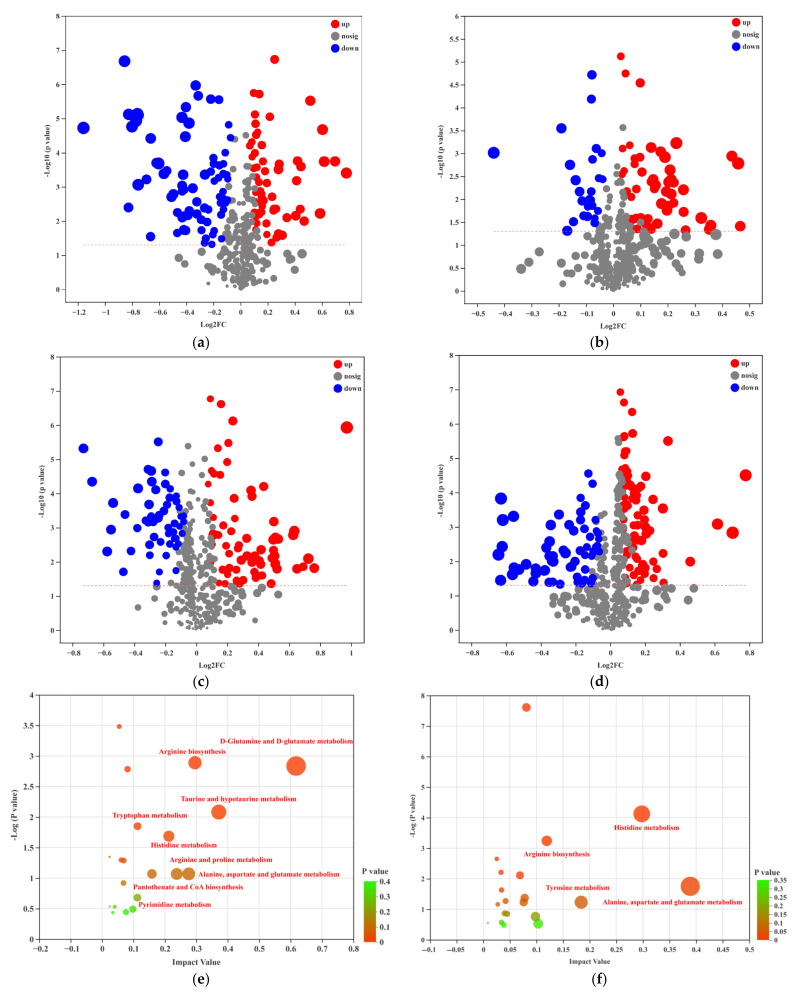
(**a**) The volcano plot of CON to MOD; (**b**) MOD to KBF-L; (**c**) MOD to KBF-M; (**d**) MOD to KBF-H. The bobble plot of pathway analysis of the metabolites improved by KBF. (**e**) CON and MOD; (**f**) MOD and KBF-L; (**g**) MOD and KBF-M; (**h**) MOD and KBF-H.

**Figure 4 nutrients-14-03202-f004:**
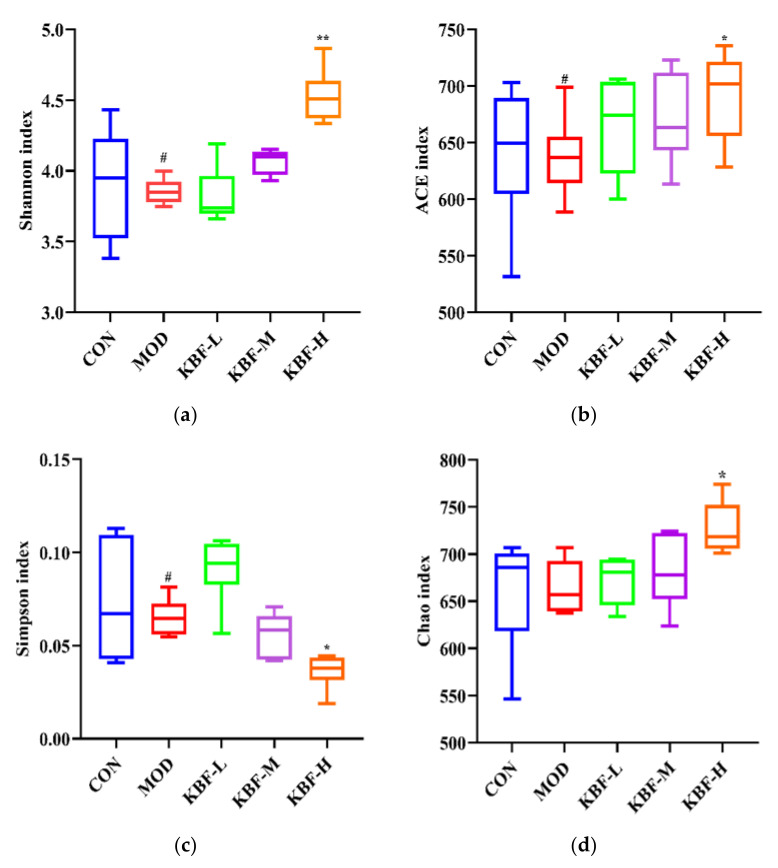
Alpha diversity indexes of (**a**) Chao; (**b**) ACE; (**c**) Shannon; (**d**) Simpson. Data are expressed as the mean ± SD (*n* = 6). (^#^
*p* < 0.05, versus CON), (* *p* < 0.05, ** *p* < 0.01 versus MOD).

**Figure 5 nutrients-14-03202-f005:**
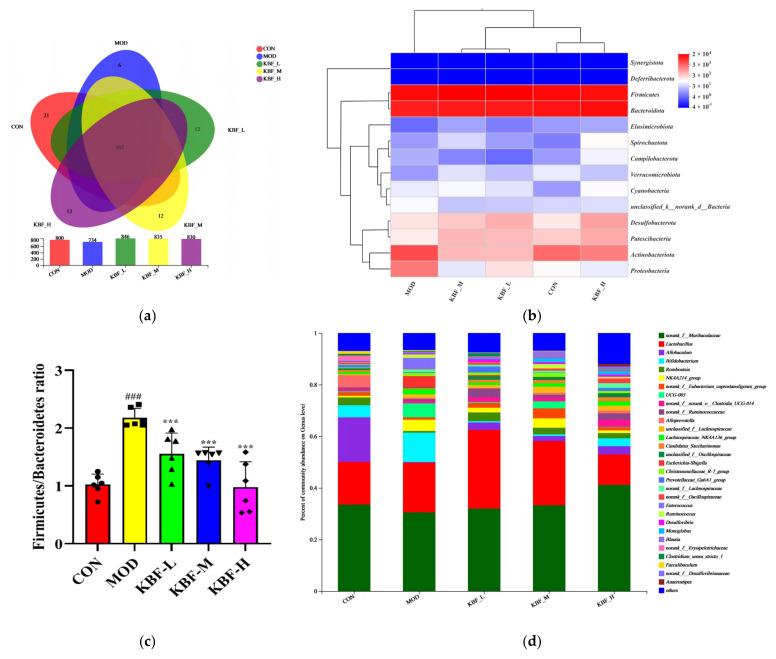
Effects of the kidney bean fermentation broth on the gut microbiota composition (**a**) Venn diagram. (**b**) Heatmaps of bacterial distribution at phylum taxonomic levels. (**c**) Ratio of Firmicutes and Bacteroidetes. (^###^
*p* < 0.001 versus CON, *** *p* < 0.001 versus MOD), (**d**) Bacterial taxonomic profiling at genus taxonomic levels.

**Figure 6 nutrients-14-03202-f006:**
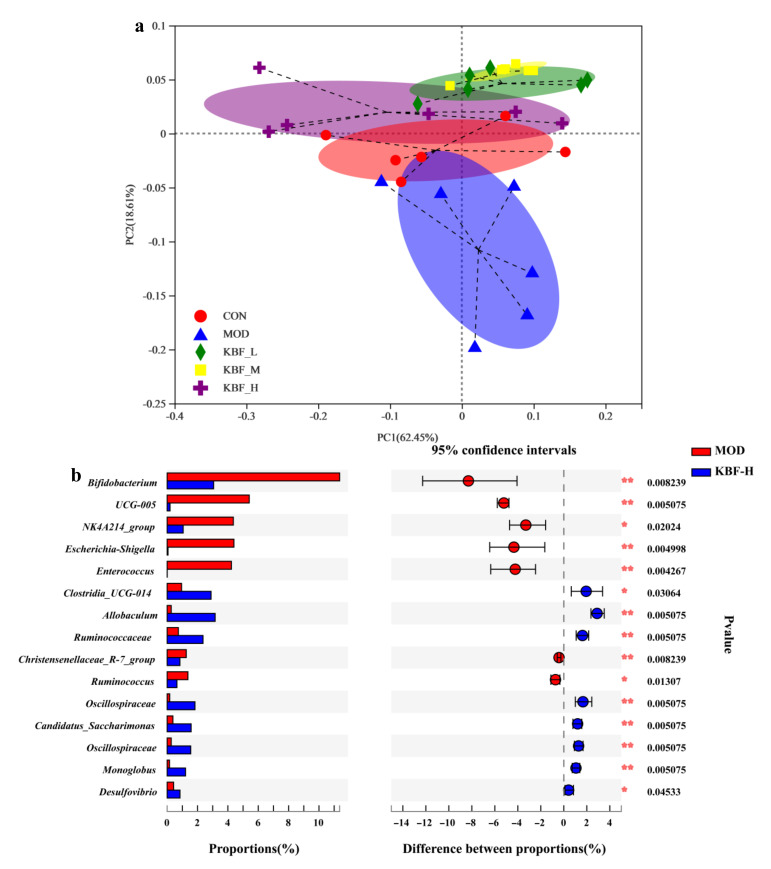
(**a**) βeta-diversity analysis based on unweighted Unifrac distance using PCoA. (**b**) The Lefse analysis at the genus level. (LDA score > 2, *p* < 0.05, FDR < 0.1), LDA results between MOD and KBF-H groups. (**c**) LDA results in all groups. (**d**) The significantly changed bacteria at the genus level based on LDA results in all groups.

**Figure 7 nutrients-14-03202-f007:**
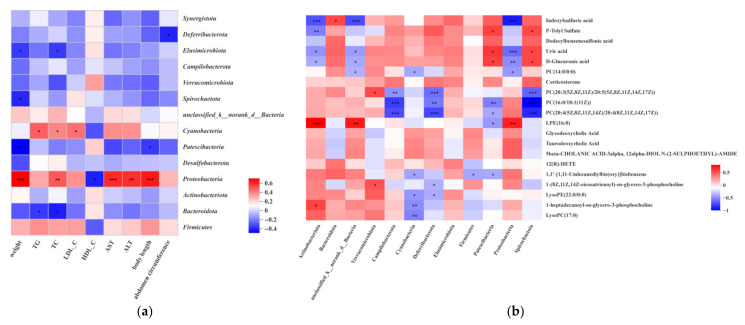
The correlation among targeted bacteria, apparent indicators, and metabolites (in serum and colon). (**a**) The correlation between apparent indicators and targeted bacteria (in colon and serum, directions of change in the top 30 genera). (**b**) The correlation between targeted bacteria and metabolites (directions of change in the top 50 metabolites). (**c**) The correlation between metabolites and apparent indicators (in colon and serum). Data were presented as mean ± SEM. Different letters indicated statistical differences between groups (* *p* < 0.05, ** *p*< 0.01 and *** *p*< 0.001).

**Table 1 nutrients-14-03202-t001:** Basic and bioactive components of KBNF and KBF.

	KBNF	KBF
Protein (g/100 g)	0.54 ± 0.03	0.45 ± 0.01 ^▲^
Ash (g/100 g)	0.48 ± 0.02	0.31 ± 0.03 ^▲▲^
Carbohydrate (g/100 g)	11.61 ± 0.02	1.82 ± 0.04 ^▲▲▲^
Starch (g/100 g)	0.11 ± 0.02	0.04 ± 0.01 ^▲▲^
Energy (kJ/100 g)	205.32 ± 0.26	40.63 ± 0.25 ^▲▲▲^
Soluble solids (%)	11.50 ± 0.05	4.60 ± 0.02 ^▲▲▲^
Total polyphenol (mg/mL)	30.02 ± 0.62	92.24 ± 2.14 ^▲▲▲^
Total flavonoid (μg/mL)	235.89 ± 2.08	367.75 ± 4.24 ^▲▲▲^

Values are presented as mean ± SD (*n* = 3) (^▲^
*p* < 0.05, ^▲▲^
*p* < 0.01 and ^▲▲▲^
*p* < 0.001 compared to KBNF).

## Data Availability

The data presented in this study are available on request from the corresponding author.

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
