# Peer review of "Kidney Bean Fermented Broth Alleviates Hyperlipidemic by Regulating Serum Metabolites and Gut Microbiota Composition"

_nutrients, 2022, doi:10.3390/nu14153202_

Round 1

Reviewer 1 Report

Methods:

48 rats --> 4 group with 6 rats each = 6x4=24 rats + 6 control... 18 rats missing... please explain

The authors describe an experiment with 48 (30) rats receiving kidney bean fermented broth in low/medium or high dose vs. unfermented broth under a high fat diet. They investigate the effects on gut microbiota and impact on fat metabolism parameters.

Results are displayed concisive and in a good order: After reporting the components of both diets under investigation, their biological effect on the rats (body weight/food intakt, abdominal circumference and effect on organs and lipid levels) is described for all groups.  Next, the impact on gut microbiota is studied accross the groups using different indices. Finally the lipid parameters and microbiota/metabolites are correlated.
The discussion takes up all major findings.

Author Response

我们要感谢审稿人抽出时间对我们的手稿发表评论。请参阅附件。 

Reviewer 2 Report

Your work provided interesting information related to an alternative method of hyperlipidemia and obesity management, high risk conditions with an increasing world-wide prevalence. Your results were very well described and the conclusions were supported by the complex analysis of your data.

Author Response

非常感谢审稿人给出的专业建议。请参阅附件。 
